# Hierarchical Analysis of Forms of Support for Employees in the Field of Health Protection and Quality of Work during the COVID-19 Pandemic and the Desired Post-Pandemic Forms of Support

**DOI:** 10.3390/ijerph192315509

**Published:** 2022-11-23

**Authors:** Izabela Dembińska, Agnieszka Barczak, Tomasz Rostkowski, Sabina Kauf, Natalia Marska-Dzioba

**Affiliations:** 1Faculty of Economics and Engineering of Transport, Maritime University of Szczecin, 70-500 Szczecin, Poland; 2Department of System Analysis and Marketing, Faculty of Economics, West Pomeranian University of Technology in Szczecin, 70-310 Szczecin, Poland; 3Human Capital Institute, Collegium of Business Administration, Warsaw School of Economics, 02-554 Warszawa, Poland; 4Department of Logistics and Marketing, Institute of Management and Quality, Opole University, 45-040 Opole, Poland; 5Institute of Economics and Finance, University of Szczecin, 70-453 Szczecin, Poland

**Keywords:** employee, COVID-19, post-pandemic, health, quality of work, Ward’s method, hierarchical method, clustering method

## Abstract

Issues of employee support during the COVID-19 pandemic and the post-pandemic period are of an interdisciplinary nature. Moreover, these should be considered from both an epistemological and a practical perspective. The aim of this study was to determine what forms of support for employees in terms of health and quality of work were provided by employers during the pandemic and what forms of support will be expected by employees after it ceases. The research process was carried out in two stages: primary and secondary exploration and quantitative clarification. In the first stage, a systematic review of the literature and a critical analysis of the so-called grey literature was performed. In the second stage, computer-assisted telephone interview (CATI) methodology was used. Ward’s method was used for data analysis. The results showed that the COVID-19 pandemic forced employers to search for new solutions to enable the continuation of their business activities, which consisted of switching from the traditional form of work to a remote form. The transition to the remote work mode changed the approach to the forms of work support provided for employees, with particular emphasis on the health of employees and the quality of work. The changes in the forms of support for employees in terms of health and quality of work were either bottom-up or top-down. Employers tried to provide access to remote infrastructure as much as possible, but the consequences of remote work in terms of the physical and mental health of employees were rarely noticed or considered. After the pandemic, online health support and access to the appropriate equipment and tools for remote work are unlikely to be needed.

## 1. Introduction

The COVID-19 pandemic has disrupted well-established practices in every aspect of economic and social life, leading to attempts to adapt to this new, unexpected situation around the world. It forced the necessity of a new look at the possible methods of running a business, opening up to less than traditional forms of the functioning of enterprises in the production and service sphere, as well as for various public entities [1,2,3,4]. In many cases, it was necessary to search for solutions such as remote work practically overnight. Remote work was introduced by employers to reduce the risk of infections but was also required by employees due to family conditions; for example, children often did not have access to educational institutions and required childcare while at home. Entrepreneurs who previously opposed the idea of remote work were forced to submit to these changes because it was a necessary adjustment to the new reality [5].

Remote work is not something new. It has been of interest to managers since the implementation of new information and communication technologies (ICT). Initially, this form of work was treated as a privilege, a luxury even; as such, it was not a popular practice [6]. The COVID-19 pandemic changed the attitudes toward remote work, which often became the only solution for many entrepreneurs, protecting them against the closure and potential collapse of their companies. However, as a result of the use of online work, many previously unknown problems appeared related to the organization of remote work as well as to the supervision and monitoring of work performance or employee support. These problems are becoming more important, especially since it is assumed that remote working and related technologies will be a driving force for enterprises and organizations in the long run [7].

A company’s performance is closely related to the performance of employees [8,9]. This is determined by a healthy work environment that enables employees to maintain their physical and mental health [10]. Working conditions have a negative or beneficial effect on the safety, health, and well-being of workers [11]. The well-being perceived by employees in relation to the workplace is closely related to the evoking of positive attitudes and behavior in employees [12]. There is no universally agreed upon standard for assessing the well-being of workers, but the general understanding of well-being refers to an individual, subjective assessment of the extent to which something contributes to improving one’s quality of life [13]. In other words, the well-being of workers refers to the idea that the quality of life is improved through the health, happiness, comfort, and peace of mind that workers experience while working [14].

The new work conditions caused by the COVID-19 pandemic have become an important research problem, giving scientists a large field of knowledge to explore. It became justified to investigate whether and how the pandemic changed the preferences and needs of employees as well as whether and how the pandemic changed the behavior of entrepreneurs towards their employees. Recognizing this research challenge, two research questions were asked: (a) What forms of support for employees in the field of health protection and quality of work were used by employers during the COVID-19 pandemic?; (b) What forms of post-pandemic support in the field of health protection and quality of work are desired by employees? Before embarking on empirical research to answer these questions, the state of knowledge in this area was reviewed.

A review of the literature allowed us to conclude that there is a research gap in relation to the research problem posed. There are studies on the transition to remote work, often analyzing the course of the transition process and its consequences for employees or employers [15,16]. As far as the consequences of the transition to remote work are concerned, the financial aspect has become an important topic, including the economic and financial impact of remote work on employees. The research of authors [17,18] has shown that most workers experience negative economic and financial impacts due to the additional costs incurred by digital technology platforms and media as well as by the lack of overtime payment and meal vouchers, which are associated with costs greater than the savings associated with fewer travel and out-of-pocket expenses. Moreover, it has been shown that psychological−behavioral variables, job satisfaction, and technostress in particular are essential in the choice to engage in further remote work after the COVID-19 pandemic. A research thread has also emerged regarding workers’ stress and anxiety about the pandemic. The COVID-19 pandemic increased the amount of uncertainty in people’s lives and was itself a source of new uncertainty. For example, studies [19,20,21] considered whether this new type of uncertainty, especially that related to the work environment, affects the level of workers’ well-being. Research has focused on five unknowns related to the pandemic and employment—the virus, quality of work, workload, work−life logistics, and employer support. It also became justified to study the psychological impact of the pandemic on employee productivity [22,23].

When looking at research directly related to the issue of supporting employees during the pandemic, one can notice that so far, there is a focus on the industry approach. The most frequent studies concerned the support of teachers’ work [24], which is justified by the scale of remote learning, or support for hospital employees [25,26] and hotels [27,28] due to the above-average risk of transmission through direct contact with patients or clients. Unfortunately, there is a deficit in employee support research during a pandemic that takes into account a multi-industry approach, allowing for comparative inferences to be made. There is also a lack of research that would answer the question of what forms of post-pandemic support in the field of health protection and work quality are desired by employees. These research gaps prove that research on employee support in the field of health protection and quality of work used by employers during the COVID-19 pandemic and in the post-pandemic period is desirable not only for epistemological reasons, but also for pragmatic reasons.

## 2. Materials and Methods

### 2.1. Context of the Research

The subject of this research is the support system implemented for workers to maintain health and quality of work during the COVID-19 pandemic and the support system desired after the pandemic. Assuming that the pandemic affected the behavior of employers towards employees, the primary objective of the research is to determine how the pandemic affected the support system of employees in terms of health care and quality of work. The assumption (thesis) for the research was made according to the logic of abduction using a deductive–empirical approach based on formal logic (deduction and the formulation of logical conclusions based on the analysis of the literature) and inductive logic (induction from direct exploratory research).

To facilitate the primary research objective, two research questions were posed:-What forms of employee health and quality of work support were used by employers during the COVID-19 pandemic?-What forms of post-pandemic support for health and quality of work are desired by workers?

It should be noted that the research problem concerns a current phenomenon with a history of only three years, which significantly determines the scarce state of knowledge in this field. This is confirmed by the literature review carried out. This deficit can be considered both in the context of the scientific problem and in the context of the research problem.

### 2.2. Stages of Scientific Reconnaissance

The course and structure of the research procedure was influenced by the nature of the objective defined at the theoretical level and was related to the lack of conceptualization and operationalization of the problem under study in the literature found to date. The research process was subordinated to abductive logic. It was conducted in two stages: the secondary and primary exploration and the quantitative explanation. The first stage was exploratory in nature. Its purpose was to justify the need for the research, to recognize the existing state of knowledge, and to identify gaps in the knowledge. In carrying out this stage, the following research techniques were used: a systematic literature review and a critical analysis of the so-called grey literature. The second stage of the research was aimed at providing empirical material to verify the validity of the adopted research thesis and to obtain answers to the research questions posed. This stage involved a computer-assisted telephone interview (CATI).

### 2.3. Participants

The target research sample was based on the indexation identified in the Polish Classification of Activities. The Polish Classification of Activities is the division of economic activities carried out by business entities in Poland. It was created for the purposes of statistics, recording, accounting, documentation, and use in official information systems and registers. It covers everyone who undertakes economic activity. It was developed on the basis of the NACE Rev2’s statistical classification of economic activities, which was introduced by Regulation (EC) No. 1893/2006 of the European Parliament and of the Council of 20 December 2006. On this basis, a population of respondents amounting to 10,130 persons was selected. The population consisted of employees of enterprises. The following attributes were determined from the population: province, occupational situation, industry, employment in the enterprise, job position, and form of employment. The entire population was invited to participate in the survey, resulting in 2430 respondents, giving an effective maneuver rate of 24%. It can be considered that from a statistical point of view, the surveyed sample is representative of the population.

### 2.4. Data Collection

Individual structured interviews were conducted between May and June 2021 via a computer-assisted telephone interview (CATI). A pilot interview was also conducted to verify the correctness of the questions. The interviews were digitally recorded and transcribed verbatim.

The institution that collected the data for us has a designated data protection officer. This person represents the data protection organization. The data protection officer was mentioned in the market survey. In addition, the survey was anonymous. All survey invitations included information about the possibility to opt out of answering questions regarding data protection. Thus, the study was conducted in accordance with Regulation (EU) 2016/679 of the European Parliament and of the Council of 27 April 2016 on the protection of individuals with regard to the processing of personal data and on the free movement of such data.

### 2.5. Data Analysis

One of the techniques belonging to the group of agglomerative methods, the Ward method, was used for this study [29]. The usefulness of Ward’s method is evidenced by the fact that it has been applied in research by many authors [30,31,32,33,34].

Agglomeration methods gather elements into clusters. Clustering is the process of grouping (hierarchizing) data into classes (groups) or clusters according to the rule that the elements in the cluster are very similar to each other while simultaneously being very dissimilar to the elements found in other clusters. Cluster methods, or hierarchical methods, create a nested sequence of partitions, with a single, all-inclusive cluster at the top and single clusters of individual objects at the bottom. Each intermediate level can be seen as a combination of two clusters from the next lower level or as a partition and a cluster from the next higher level. The results of the hierarchical clustering algorithm may be represented graphically in the form of a tree called a dendrogram. This tree graphically represents the process of merging and intermediate clustering. It thus shows how the individual points merge into a single cluster [35].

Clustering methods are divided into two groups: agglomerative and partition methods. This classification depends on whether hierarchical decomposition is created in a bottom-up or top-down manner [36]. Agglomerative clustering combines members into clusters, and these clusters continue to merge into larger clusters to form a hierarchy of clusters. Partition clustering involves constructing different partitions, which are then evaluated according to certain criteria [37,38,39,40,41,42,43]. Thus, the rationale for using Ward’s method is to combine elements into clusters so that the variance in the clusters is minimized.

As mentioned before, Ward’s method is hierarchical, i.e., it divides the elements into a dedicated number of clusters. The clustering process takes place in stages. At the beginning, each element is independent, and then, step by step, successive elements are assigned to the cluster. In each step, the elements that are the “closest” to the identified clusters are taken into account. The number of steps may be from 1 to n, where n is the number of items analyzed. For 1, only one cluster contains all members, and for n, all members form their own cluster. It should be noted that once a cluster has been created, the elements of the new cluster cannot be split again [35].

For the purposes of our study, based on the methodology used by the authors [34,44], the Ward method was carried out according to the following steps:Normalization is usually applied because of the possible scale differences among the questions (*j*):
(1)$$m_{k,j\;}=\frac{x_{k,j}-x_{mn,j}}{s}$$
where:


$x_{k,\;j}$—answer value (*x*) of the questions (*j*) concerning the users (*k*);$x_{mj}$—mean of the answers (*mn*) regarding a question (*j*);$s_{j}$—standard deviance (*s*) regarding a question (*j*);$m_{k,j}$—normalized answer value (*m*).


2.Designation of the distance (*d*) between two users or clusters (*k* and *l*) was calculated with the quadratic Euclidean distance using the normalized values for the total number of questions (*q*):


(2)
$$d\left(k,l\right)=\sum_{j=1}^{q}{\left(m_{kj}-m_{lj}\right)}^{2}$$


3.Users or clusters of minimal distance to each other are unified into a new cluster (*k* + *l*). If the new cluster exists, its distances have to be redefined towards all other users or clusters (*a*). Different clustering methods use different algorithms for the calculation of new distances. The Ward method calculates the optimal minimum distance by taking into account the number of users in the clusters.

(3)$$d\left(a,\;k+l\right)=\frac{\left(N_{a}+N_{k}\right)*d\left(a,\;k\right)+\left(N_{a}+N_{l}\right)*d\left(a,\;l\right)-N_{a}*d\left(k,\;l\right)}{N_{a}+N_{k}+N_{l}}$$
where:


$N_{a}$—number of users (*a*) in the cluster;$N_{k}$—number of users (*k*) in the cluster;$N_{l}$—number of users (*l*) in the cluster.


The horizontal lines mark the stage at which the grouping was interrupted. The intersection was determined based on the analysis of the bonding distance in relation to the bonding steps (Euclidean distance).

## 3. Results

For the purposes of this study, 26 variables were considered regarding non-wage forms of employee support during the COVID-19 pandemic and expected support after its termination. Both offered and expected services were analyzed in terms of gender, age, size of the city in which the respondents live, the voivodeship in which the respondents live, the work situation they are in, the amount of employment in the company, the industry in which the organization operates, the position in which the respondent is employed, and the type of employment contract (70 variables). A voivodeship is the highest-level administrative division in Poland. We were interested in whether there would be variation in the responses by voivodeship. In Poland, there are differences between so-called northern and southern as well as western and eastern Poland, similar to in Germany or Italy. As the variables gender, age, and size of the town/city in which the respondents live did not show any correlation with regard to the subject of this study, they were omitted. The variables used in the analysis are presented in Table 1 and Table 2.

Before starting the analysis, it was examined whether the variables were dependent. As the responses obtained are measured on a nominal scale, Pearson’s $\chi^{2}$ test of independence was used to assess the dependency of the variables. All of the variables presented in Table 1 and Table 2 met the required dependency conditions and thus could be used for further analysis.

For the sake of clarity, the graphs were divided into separate issues in order to illustrate the relationship between the categories of variables more easily. First, the relationships between the respondents’ characteristics and non-wage forms of support offered by employers during the COVID-19 pandemic are presented (Figure 1, Figure 2, Figure 3, Figure 4, Figure 5 and Figure 6).

The following clusters were obtained, on the basis of which regularities regarding the P2 variable were identified (Figure 1). Cluster 1: Inhabitants of the Holly Cros Voivodeship (P2: 13) indicated that their employers cared about the education of employees by sending them to workshops and training sessions as well as to seminars and webinars (P15_5: 1). Cluster 2: Employees from the Subcarpathia Voivodeship (P2: 9) claimed that their employers asked employees about their actual health care needs and ways to effectively use the benefits offered (P15_1: 1). Cluster 3: Inhabitants of the Warmia-Masuria Voivodeship (P2: 14) believe that their employers conducted employee satisfaction surveys regarding individual benefits (P15_6: 1). The respondents from this voivodeship were rather willing to admit that employers provided them with access to appropriate equipment and tools to enable remote work (P15_8: 1). Cluster 4: Employees from the Lubusz Voivodeship (P2: 4) believed that employers provided them with health and safety advice related to COVID-19 (P15_4: 1). They were also willing to admit that their employers took care to ensure safety while performing remote work (P15_7: 1). Cluster 5: Inhabitants of the Lesser Poland Voivodeship (P2: 6) were of the opinion that employers their did not offer them individual counseling regarding the selection of benefits to meet the needs of employees (P15_3: 0). They were also willing to admit that they did not receive any forms of support other than those indicated in the questionnaire (P15_13: 0) and that their employers did not provide the security of remote work (P15_7: 0). Cluster 6: Inhabitants of the Silesia Voivodeship (P2: 12) were of the opinion that their employers did not provide them with adequate online support regarding healthy eating (P15_12: 0) and that they did not conduct research on employee satisfaction with individual benefits (P15_6: 0). Employees from the Kuyavia-Pomerania Voivodeship (P2: 2) and the Lublin Voivodeship (P2: 3) also had similar opinions on this subject.

The following clusters were obtained, on the basis of which regularities concerning the P3 variable were identified (Figure 2). Cluster 1: Full-time employed persons (P3: 8) believed that their employers did not provide them with the security of remote work (P15_7: 0) and did not provide online support in the field of physical activity (P15_11: 0). Cluster 2: Self-employed persons (freelancers) (P3: 4) believed that their employers did not provide them with online support regarding taking care of their health (P15_10: 0) and eating healthy (P15_12: 0). 

The following clusters were obtained, from which regularities were identified for variable P4 (Figure 3). Cluster 1: According to respondents in companies with 1 to 9 employees (P4: 1), employers did not conduct surveys on employee satisfaction with individual benefits (P15_6: 0) and did not offer online health support (P15_10: 0). Employees in this cluster were also inclined to say that their employers were unlikely to offer individual counselling regarding the selection of benefits to meet the needs of the employee (P15_3: 0) and did not offer counselling regarding health security related to COVID-19 (P15_4: 0). Employees in this group were unlikely to use forms of support other than those mentioned in the survey (P15_13: 0). Cluster 2: Employers with 10 to 49 employees (P4: 2) did not offer their employees online support regarding healthy eating (P15_12: 0). Cluster 3: According to the respondents, employers with 150 to 249 employees (P4: 3) did not conduct employee satisfaction surveys on individual benefits (P15_6: 0) and did not offer them online health support (P15_10: 0). Cluster 4: According to the respondents, employers with more than 500 workers (P4: 5) offered their employees advice on safety related to COVID-19 (P15_4: 1).

The following clusters were obtained, from which regularities were identified for variable P5 (Figure 4). Cluster 1: Workers employed in mining and quarrying (P5: 2) believed that their employers did not offer them COVID-19 health safety advice (P15_4: 0) and did not provide them with online health support (P15_10: 0) or other forms of support apart from those mentioned in this study (P15_13: 0). Cluster 2: Those employed in information and communication: media, publishing houses, and ICT services (P5: 10), believed that their employers cared about the education of employees (P15_5: 1) and ensured the security of remote work (P15_7: 1). Cluster 3: Employees in construction (P5: 6) and public administration and national defense, including compulsory social and health security (P5: 15), believed that their employers did not conduct research on the real needs of employees in terms of health care and services offered (P15_1: 0). Cluster 4: Employees working in transport and warehouse management (P5: 8) believed that their employers did not measure the level of satisfaction with individual benefits (P15_6: 0) and that they did not offer online support for employees regarding healthy eating (P15_12: 0). Cluster 5: Those employed in wholesale and retail trade and motor vehicle repair (P5: 7); professional, scientific, and technical entities (P5: 13); and arts, entertainment, and recreation and sports (P5: 18) believed that their employers did not provide them with access to healthy food at the company and at home offices (P15_9: 0). Cluster 6: Employees employed in other sectors of the national economy than those indicated in the study (P5: 20) believed that their employers did not ensure their safety when working remotely (P15_7: 0).

The following clusters were obtained, from which regularities were identified for variable P6 (Figure 5). Cluster 1: Employed individuals (P6: 1) believed that their employers did not provide them with safe conditions during remote work (P15_7: 0) and did not provide them with access to appropriate equipment and tools enabling them to perform work in this way (P15_8: 0). They were also willing to admit that their employers did not offer them health and safety advice related to COVID-19 (P15_4: 0). Cluster 2: People employed as a specialist (P6: 2) believed that they did not receive online support from their employers regarding healthy eating (P15_12: 0). Cluster 3: Both lower- (P6: 3) and higher-level managers and executives (P6: 4) acknowledged that they have received COVID-19 health and safety advice from their employer (P15_4: 1).

The following clusters were obtained, from which regularities were identified for variable P7 (Figure 6). Cluster 1: Those working under a fixed-term employment contract (P7: 1) believed that their employers did not test employee satisfaction with individual benefits (P15_6: 0) and did not provide them with the security of working remotely (P15_7: 0). Employees in this group were rather willing to admit that their employers did not provide them with access to healthy food at the company and at home offices (P15_9: 0). Cluster 2: Employees with permanent contracts (P7: 2) were not able to receive individual counselling on the selection of benefits to meet employee needs (P15_3: 0). They were more likely to admit that they did not receive any other benefits from their employers other than those indicated on the survey form (P15_13: 0). Cluster 3: Contracted employees (P7: 3) are more likely to admit that their employers did not provide them with online physical activity support (P15_11: 0). Cluster 4: Persons employed on the basis of a mandate contract (P7: 4) believed that their employers did not provide them with online support for physical activity (P15_11: 0). Cluster 5: People employed under a b2b contract (P7: 5) believed that their employers offered them other forms of support than those indicated in the survey (P15_13: 1). Cluster 6: Those employed in a way other than indicated in the study (P7: 6) believed that their employers did not care for the education of employees (P15_5: 0) and that they did not provide them with access to appropriate equipment and tools to enable remote working (P15_8: 0).

The next stage of the study was to analyze the relationships between the respondents’ characteristics and the non-wage forms of support that employees would like to receive from their employers (Figure 7, Figure 8, Figure 9, Figure 10, Figure 11 and Figure 12).

The following clusters were obtained, from which regularities were identified for variable P2 (Figure 7). Cluster 1: People from the Lower Silesia Voivodeship (P2: 1) are not interested in future employers examining the actual needs of employees regarding the expected forms of support (P16_1: 0). According to this group of respondents, they will not need counselling on health care and preventive health care in the future (P16_2: 0). Cluster 2: Inhabitants of the Kuyavia-Pomerania Voivodeship (P2: 2) believe that there is no need for employers to test their satisfaction with particular benefits in the future (P16_6: 0). They are willing to admit that they will not need individual counseling on how to select benefits to meet their needs (P16_3: 0) and that they are unlikely to be interested in the forms of employee education offered by employers (P16_5: 0). Cluster 3: Employees from the Lublin Voivodeship (P2: 3) believe that they will not want to use online support for healthy eating in the future (P16_12: 0). Cluster 4: People from the Lodzkie Voivodeship (P2: 5) will not want to use other forms of support than those indicated in the survey in the future (P16_13: 0). They are inclined to say that they are unlikely to need online support in terms of physical activity (P16_11: 0). Cluster 5: Employees from the Pomeranian Voivodeship (P2: 11) say that they will not use online support for physical activity in the future (P16_11: 0). Furthermore, they claim that they are unlikely to use other forms of support than those indicated in the study (P16_13: 0). Cluster 6: People from the Masovian Voivodeship (P2: 7) and the Greater Poland Voivodeship (P2: 15) believe that in the future, they will not be interested in the employer providing access to appropriate equipment and tools that enable remote working (P16_8: 0). Respondents from these provinces will also not be interested in the employer providing security for remote work (P16_7: 0). In their opinion, the provision of COVID-19-related health and safety advice by their employers is unlikely to be needed anymore (P16_4: 0).

The following clusters were obtained, from which regularities were identified for variable P3 (Figure 8). Cluster 1: Respondents with full-time employment (P3: 8) believe that in the future, it will not be necessary for the employer to provide online support for healthy eating (P16_12: 0) as well as online support for health protection (P16_10: 0) and that there will be no need for employers to provide security for remote work (P16_7: 0). People in this group are inclined to say that there is unlikely to be a need for individual counselling to match benefits to employee needs (P16_3: 0).

The following clusters were obtained, from which regularities were identified for variable P4 (Figure 9). Cluster 1: Respondents working in companies with 1 to 9 employees (P4: 1) believe that after the pandemic, they will not need individual counselling to match benefits to employee needs (P16_3: 0). Cluster 2: Those working in companies with 10 to 49 employees (P4: 2) will not be interested in other forms of support than those indicated in the questionnaire (P16_13: 0). Cluster 3: In companies with more than 500 employees (P4: 5), employees see no need for employers to measure employee satisfaction with specific benefits in the future (P16_6: 0). In their opinion, there will also be no need to ensure the security of remote work (P16_7: 0). People in this group are inclined to say that they are unlikely to need online support for healthy eating (P16_12: 0).

The following clusters were obtained, on the basis of which regularities regarding the P5 variable were identified (Figure 10). Cluster 1: Those employed in agriculture, forestry, hunting, and fishing (P5: 1) believe that online physical activity support (P16_11: 0) will be unnecessary for them in the future. Cluster 2: Those employed in professional, scientific, and technical entities (P5: 13) believe that they are unlikely to need online physical activity support (P16_11: 0) in the future. Cluster 3: Employees in the education sector (P5: 16) believe that they are unlikely to need advice in the future regarding health safety related to COVID-19 (P16_4: 0) nor will they require access to healthy food in the company or at home offices (P16_9: 0). Cluster 4: Employees of hotels, restaurants and catering, and accommodation businesses (P5: 9) are unlikely to need other forms of support in the future than those indicated in the survey (P16_13: 0). Cluster 5: Employees of financial and insurance businesses (P5: 11) will not need any other forms of support than those indicated in the questionnaire (P16_13: 0). Cluster 6: Those employed in the wholesale and retail trade as well as in motor vehicles repair (P5: 7) believe that they are unlikely to need individual counselling regarding the selection of benefits to meet the needs of employees (P16_3: 0) or for employers to survey the level of employee satisfaction with individual benefits (P16_6: 0). In the future, employees in this group are unlikely to need individual counselling on health care and preventive health care (P16_2: 0). Cluster 7: People employed in industries other than those mentioned in the study (P5: 20) indicate that they will not need online support for healthy eating after the pandemic (P16_12: 0).

The following clusters were obtained, from which regularities were identified for variable P6 (Figure 11). Cluster 1: Employees employed as specialists (P6: 2) indicate that in the future, they will not need their employers to provide them with access to healthy food in the company and or at home offices (P16_9: 0). They believe that it is unlikely that employers should need to measure the level of employee satisfaction with particular benefits (P16_6: 0) and provide them with online support for healthy eating (P16_12: 0).

The following clusters were obtained, on the basis of which regularities regarding the P7 variable were identified (Figure 12). Cluster 1: Respondents with a fixed-term contract (P7: 1) will not be interested in individual counselling regarding health care and prevention in the future (P16_2: 0). Cluster 2: Those employed on a permanent contract (P7: 2) will not need online support for physical activity (P16_11: 0). In their opinion, online health support (P16_10: 0) and the provision of access to appropriate equipment and tools enabling remote work (P16_8: 0) will not be needed after the pandemic.

## 4. Conclusions

### 4.1. Conclusions with Commentary

The issue of supporting workers’ health and quality of work prior to a pandemic has quite often been considered within the framework of social responsibility [45,46,47,48]. Nowadays, another strand of research has emerged, in which the issue of health and quality support for workers is referred to as pandemic conditions.

The aim of the present research was to determine how the pandemic affected employee health and the quality of the support system during the COVID-19 pandemic and in the post-pandemic period. The results of the study allow the following conclusions:

The COVID-19 pandemic forced employers to introduce new solutions to their practices by switching from the traditional form of work to a remote mode. Not surprisingly, this was the only sensible solution, as lockdowns in the early phases of the pandemic were in force in all countries, and companies could not or would not shut down.

The shift to remote mode consequently forced a redefinition of the forms of work support for employees. Due to the nature of the pandemic, particular attention was paid to health and quality of work, which may be due to the fact that these are key determinants of employee productivity and efficiency, although it may also be due to social rationale, i.e., the desire to take care of one’s employee.

The change in the forms of support for employee health and quality of work was either bottom-up, when employees took the initiative, or top-down, when the initiative was on the employer’s side.

While employers tried to provide access to remote infrastructure where possible, the consequences of remote work on employees’ physical and mental health were rarely noticed or considered by them. This may result from the lack of knowledge on how to behave in this case and the lack of substantive preparation or lack of access to specialists.

Considering the form of employment, full-time employees stated that their employers did not provide them with safe conditions for remote work and did not provide online support for physical activity—this may be due to the fact that employers were surprised by the pandemic as well as by its scale and duration and did not have time to prepare for such a situation; in other words, they did not have the appropriate experience in this field. They did not have sufficient knowledge of how they could ensure safe conditions during periods of remote work. Additionally, the problem of physical activity in remote working conditions has become a society-wide issue (it is also intensely discussed in the case of distance learning). It is reasonable to assume that the approaches of employers could improve over time.

Large agglomerations tend to have business entities with a greater financial capacity that can be directed towards non-wage benefits for employees. Additionally, foreign capital may influence a different organizational culture that places more emphasis on taking care of employees during and immediately after a pandemic.

Employees of hotels, restaurants, and other catering and accommodation services believe that their employers did not provide them with adequate equipment and tools to enable them to work remotely. Such a response may be a consequence of the total lockdown that affected these industries. At that time, hotels and restaurants were completely closed, so employers saw no need to equip their employees with remote infrastructure.

Different approaches to employee support can be seen for the “company size” variable. Only employers with more than 500 employees offered their employees safety advice related to COVID-19. These results are very similar to the results of research on corporate social responsibility practices [45,49,50,51], which showed that small and medium-sized enterprises are not interested in corporate social responsibility (CSR), mainly due to the need for additional costs or time. It can be assumed that these two premises became a destructive factor in this case.

From the perspective of the position, both lower- and higher-level managers were offered COVID-19 health safety advice by their employer.

Research shows that online health support and access to appropriate equipment and tools to enable remote work after a pandemic are unlikely to be needed.

### 4.2. Research Limitations

On the one hand, Ward’s method is conservative and even monotonic, producing more or less the same large groups, but on the other hand, it is sensitive to outliers. Compared to other methods, Ward’s approach ensures greater accuracy of the results and minimizes the variance between the elements. Hands and Everitt [43] showed that in a set of five grouping techniques, Ward’s method performed better than other hierarchical methods. By running simulations on multiple datasets, Blashfield [52] came to similar conclusions.

The disadvantage of the hierarchical method is that once a merge or split is performed, it can never be undone. However, this kind of inflexibility is useful because since one does not have to worry about a combinatorial number of different options, it implies a lower computational cost.

Another limitation of using the hierarchical method is that erroneous decisions in the grouping process that have already been made cannot be corrected. The solution to this problem is to perform a careful analysis of the member associations with each hierarchical partition. It is also possible to integrate hierarchical agglomeration with other approaches by first using the hierarchical agglomeration algorithm to group elements into sets of micro-clusters and then performing macro-clustering on micro-clusters using another grouping method, such as iterative relocation [53].

### 4.3. Future Research Directions

The problem of forms of support for employee health and quality of work during and after a pandemic is a large field of research. It makes sense to expand research to extend to other industries and activities. Cross-national comparisons are also warranted. There is also a question that is worth looking at in the future: whether the time perspective will change the preferences related to forms of employee support for health and quality of work, i.e., how long the pandemic will act as a determinant in this respect.

## Figures and Tables

**Figure 1 ijerph-19-15509-f001:**
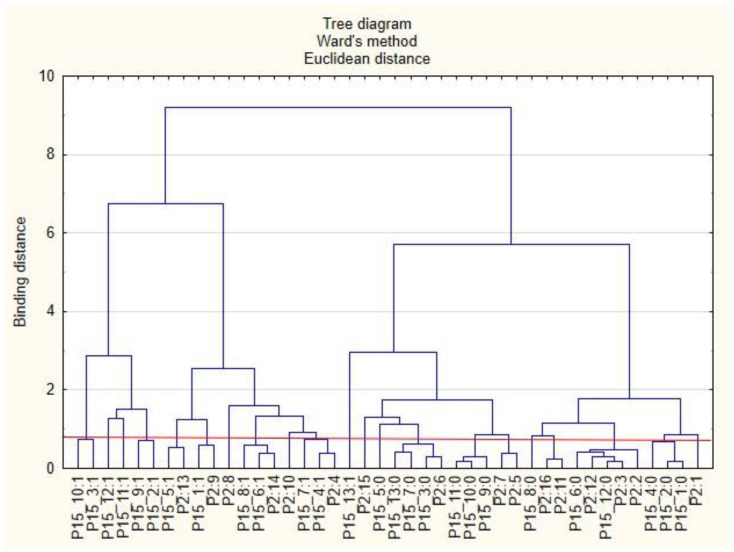
Diagram of hierarchical classification of variable categories: support received during the COVID-19 pandemic from the employer in the area of health care and the effective use of the benefits offered (P15) as well as the voivodeships where the respondents live (P2).

**Figure 2 ijerph-19-15509-f002:**
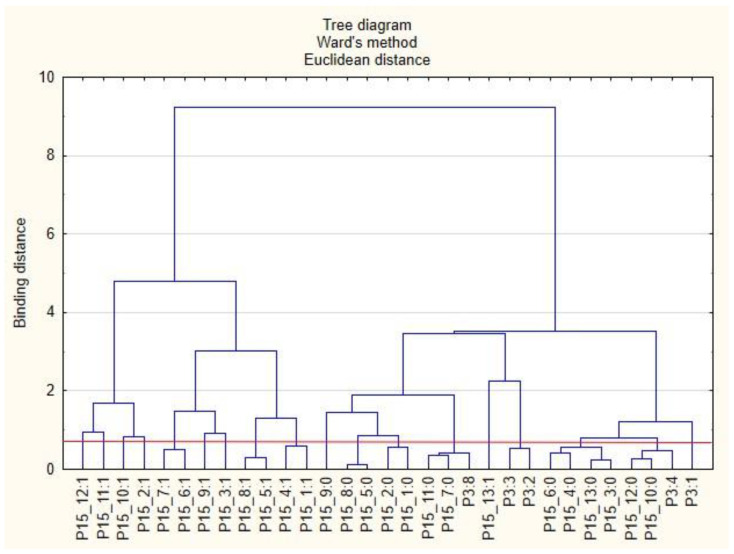
Diagram of hierarchical classification of variable categories: support received during the COVID-19 pandemic from the employer in the area of health care and the effective use of the benefits offered (P15) as well as the professional situation of the respondents (P3).

**Figure 3 ijerph-19-15509-f003:**
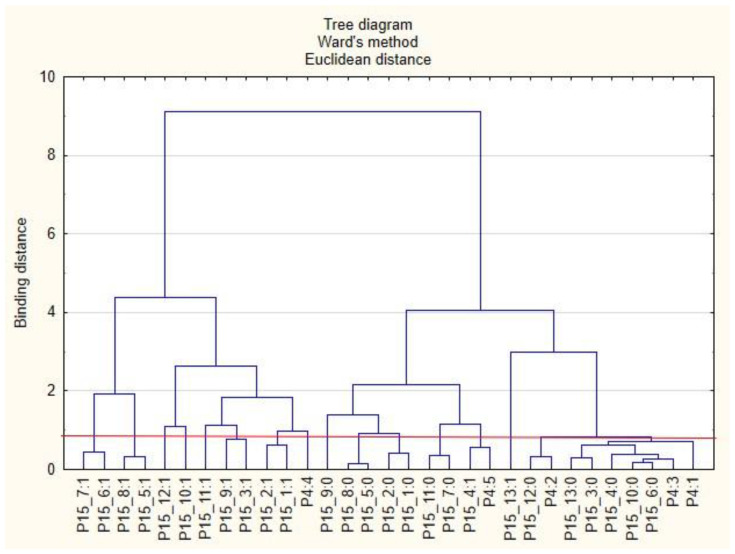
Diagram of hierarchical classification of variable categories: support received during the COVID-19 pandemic from the employer in the area of health care and effective use of the benefits offered (P15) as well as the amount of employment in the company (P4).

**Figure 4 ijerph-19-15509-f004:**
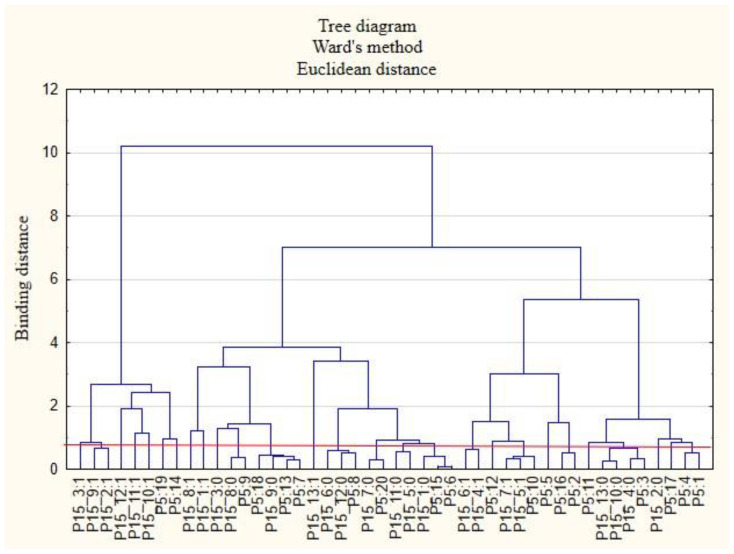
Diagram of hierarchical classification of variable categories: support received during the COVID-19 pandemic from the employer in the area of health care and the effective use of the benefits offered (P15) as well as the industries in which the respondents work (P5).

**Figure 5 ijerph-19-15509-f005:**
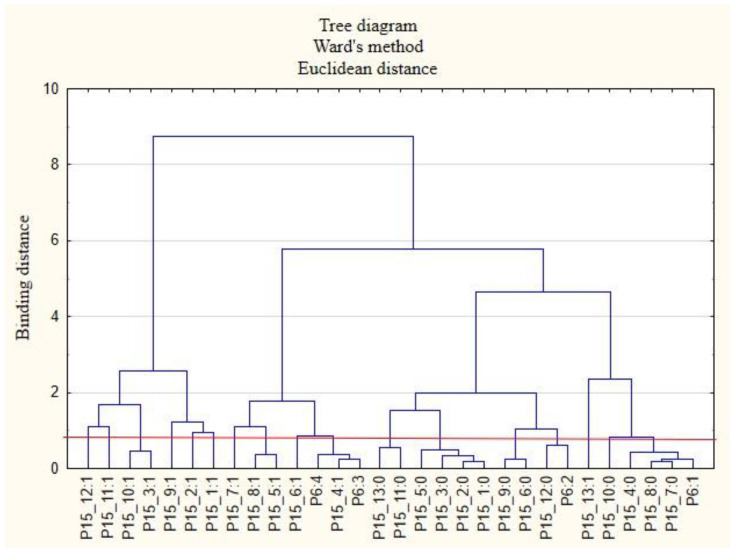
Diagram of hierarchical classification of variable categories: support received during the COVID-19 pandemic from the employer in the area of healthcare and the effective use of the benefits offered (P15) as well as the positions in which the respondents work (P6).

**Figure 6 ijerph-19-15509-f006:**
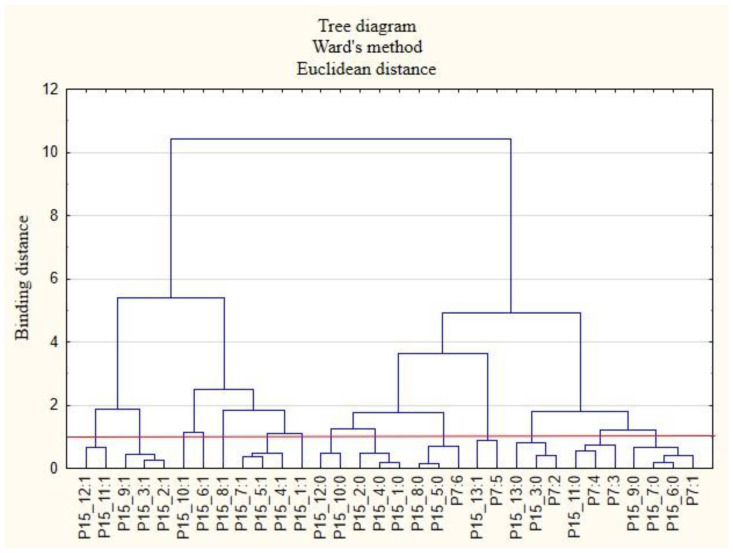
Diagram of hierarchical classification of variable categories support received during the COVID-19 pandemic from the employer in the area of healthcare and effective use of the benefits offered (P15) as well as the forms of employment of the respondents (P7).

**Figure 7 ijerph-19-15509-f007:**
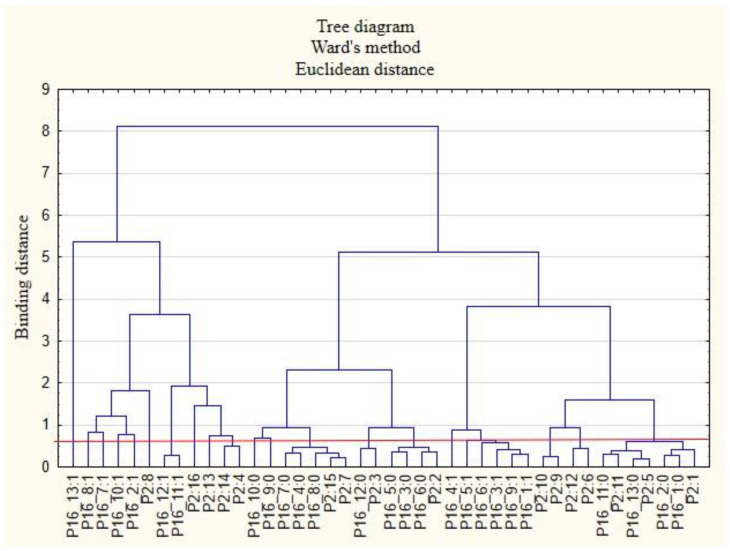
Diagram of hierarchical classification of variable categories: support that respondents want to receive after the end of the COVID-19 pandemic from the employer in the area of health care and the effective use of the benefits offered (P16) as well as the voivodeships where the respondents live (P2).

**Figure 8 ijerph-19-15509-f008:**
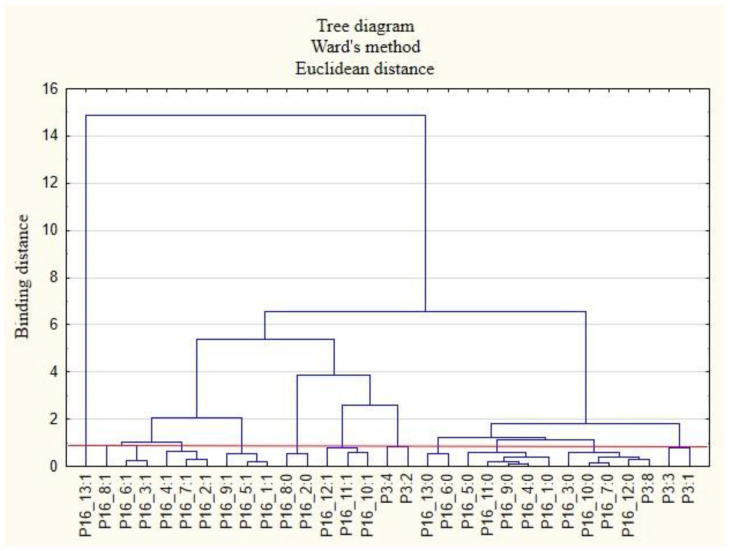
Diagram of hierarchical classification of variable categories: support that respondents want to receive after the end of the COVID-19 pandemic from the employer in the area of healthcare and the effective use of the benefits offered (P16) as well as the professional situations of the respondents (P3).

**Figure 9 ijerph-19-15509-f009:**
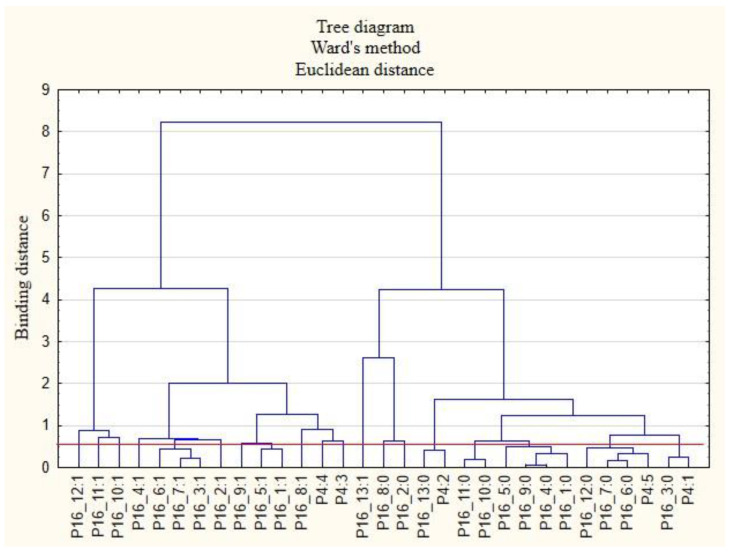
Diagram of hierarchical classification of variable categories: support that respondents want to receive after the end of the COVID-19 pandemic from the employer in the area of healthcare and the effective use of the benefits offered (P16) as well as the amount of employment in the company (P4).

**Figure 10 ijerph-19-15509-f010:**
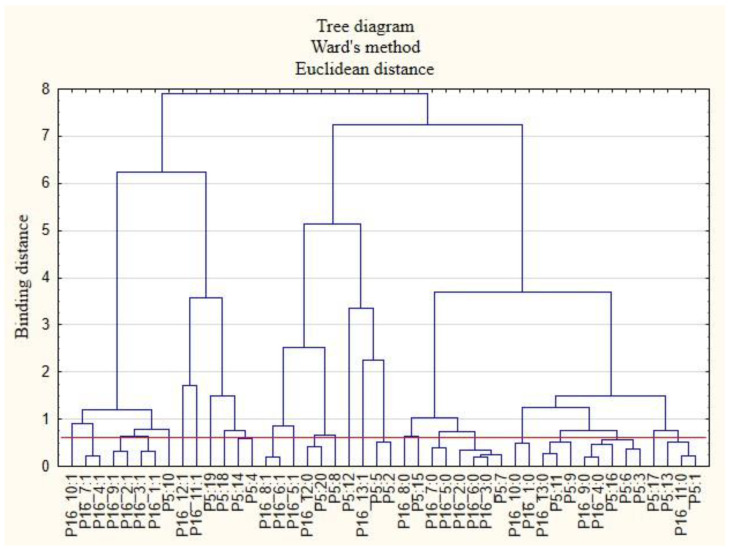
Diagram of hierarchical classification of variable categories: support that respondents want to receive after the end of the COVID-19 pandemic from the employer in the area of healthcare and the effective use of the benefits offered (P16) as well as the industries in which the respondents work (P5).

**Figure 11 ijerph-19-15509-f011:**
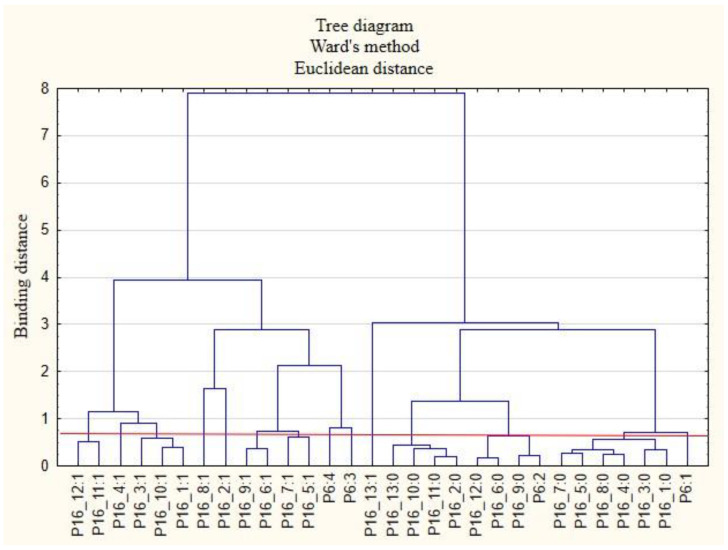
Diagram of hierarchical classification of variable categories: support that respondents want to receive after the end of the COVID-19 pandemic from the employer in the area of healthcare and the effective use of the benefits offered (P16) as well as the positions the respondents work in (P6).

**Figure 12 ijerph-19-15509-f012:**
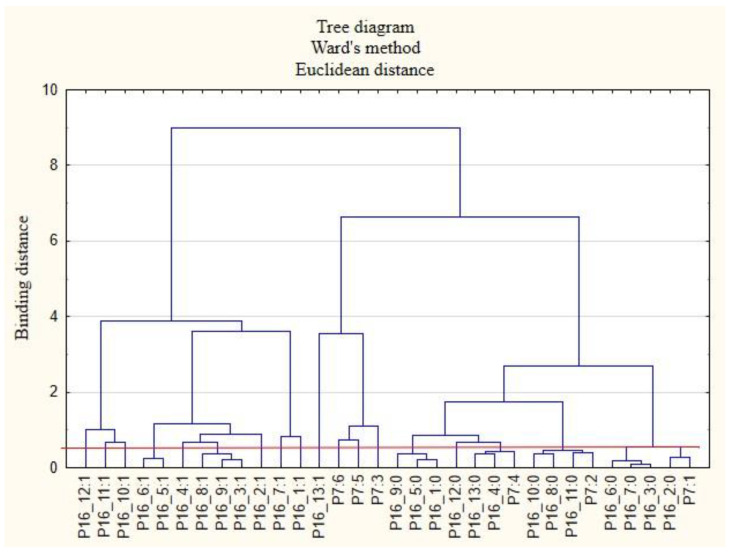
Diagram of hierarchical classification of variable categories: support that respondents want to receive after the end of the COVID-19 pandemic from the employer in the area of health care and the effective use of the benefits offered (P16) as well as the forms of employment of the respondents (P7).

**Table 1 ijerph-19-15509-t001:** Variables used in the analysis—received and desired support (0—no, 1—yes).

P15	Support received during the COVID-19 pandemic from the employer in the area of healthcare and the effective use of the benefits offered
P15_1	Researching the real needs of employees
P15_2	Individual consultancy in the field of health protection and preventive health care
P15_3	Individual counseling in the selection of benefits for employee needs
P15_4	COVID-19 health safety consultancy
P15_5	Employee education (workshops, trainings, seminars, webinars)
P15_6	Employee satisfaction survey with individual benefits
P15_7	Ensuring the security of remote work
P15_8	Providing access to appropriate equipment and tools to perform work remotely
P15_9	Providing access to healthy food in the company and at home offices
P15_10	Online health support
P15_11	Online support in the field of physical activity
P15_12	Online support for healthy eating
P15_13	Other forms of support
P16	Support that respondents want to receive after the end of the COVID-19 pandemic from the employer in the area of health care and the effective use of the benefits offered
P16_1	Researching the real needs of employees
P16_2	Individual consultancy in the field of health protection and preventive health care
P16_3	Individual counseling in the selection of benefits for the employee’s needs
P16_4	COVID-19 health safety consultancy
P16_5	Employee education (workshops, trainings, seminars, webinars)
P16_6	Employee satisfaction survey with individual benefits
P16_7	Ensuring the security of remote work
P16_8	Providing access to appropriate equipment and tools to perform work remotely
P16_9	Providing access to healthy food in the company and at home offices
P16_10	Online health support
P16_11	Online support in the field of physical activity
P16_12	Online support for healthy eating
P16_13	Other forms of support

**Table 2 ijerph-19-15509-t002:** Variables used in the analysis—characteristics of the respondents.

P2—voivodeship	P2:1	Lower Silesia
	P2:2	Kuyavia-Pomerania
	P2:3	Lublin
	P2:4	Lubusz
	P2:5	Lodzkie
	P2:6	Masovian
	P2:7	Lesser Poland
	P2:8	Opole
	P2:9	Subcarpathia
	P2:10	Podlasie
	P2:11	Pomeranian
	P2:12	Silesian
	P2:13	Holly Cros
	P2:14	Warmia-Masuria
	P2:15	Greater Poland
	P2:16	West Pomeranian
P3—professional situation	P3:1	I work part time
	P3:2	I am a self-employed full-time employee
	P3:3	I am employed but currently on long-term leave/maternity/parental leave
	P3:4	I am a self-employed person (freelancer)
	P3:5	I am learning
	P3:6	I am retired or on a pension
	P3:7	I am unemployed
	P3:8	I work full time
P4—employment in the company	P4:1	1–9 people
	P4:2	10–49 people
	P4:3	50–249 people
	P4:4	250–499 people
	P4:5	Over 500 people
P5—industry	P5:1	Agriculture, forestry, hunting, and fishing
	P5:2	Mining and quarrying
	P5:3	Industrial processing
	P5:4	Production and supply of electricity, gas, steam, hot water, and air for air conditioning systems
	P5:5	Water supply: sewerage, waste management, and remediation activities
	P5:6	Construction
	P5:7	Wholesale and retail trade; repair of motor vehicles
	P5:8	Transport and warehouse management
	P5:9	Hotels and restaurants, food services, and accommodation
	P5:10	Information and communication—media, publishing houses, ICT services
	P5:11	Financial and insurance activities
	P5:12	Real estate market services
	P5:13	Professional, scientific, and technical activity
	P5:14	Administration services and supporting activities—not requiring specialist knowledge
	P5:15	Public administration and national defense; compulsory social and health security
	P5:16	Education
	P5:17	Healthcare and social assistance
	P5:18	Activities related to culture, entertainment, recreation, and sports
	P5:19	Households with employees
	P5:20	Other
P6—workplace	P6:1	Regular employee
	P6:2	Expert
	P6:3	Junior manager/manager
	P6:4	Senior manager/manager
P7—form of employment	P7:1	Contract of employment for a specified amount of time
	P7:2	Employment contract for an indefinite period
	P7:3	Contract work
	P7:4	Contract of mandate
	P7:5	B2B contract
	P7:6	A different type of contract

## Data Availability

The study was conducted in accordance with Regulation (EU) 2016/679 of the European Parliament and of the Council of 27 April 2016 on the protection of individuals with regard to the processing of personal data and on the free movement of such data.

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
