# Peer review of "Hierarchical Analysis of Forms of Support for Employees in the Field of Health Protection and Quality of Work during the COVID-19 Pandemic and the Desired Post-Pandemic Forms of Support"

_ijerph, 2022, doi:10.3390/ijerph192315509_

Round 1

Reviewer 1 Report

Review for ijerph-1975161

What is the main question addressed by the research?

The aim of the study is to investigate what forms of employees' support were expected and provided by employers during 19 the pandemic (in terms of health care and quality of work)

There are some suggestions to improve the text:

Ln 18. The aim of the study is to indicate what forms…  Instead of "indicate" use word to explore… or to investigate… to determine…  this is more acceptable in writing a scientific paper.

Ln 117-118 the aim was described somewhat better (the aim of the research is to determine how the pandemic affected the employee support system in terms of health protection and quality of work.) so suggestion is to put this formulation in Abstract and more detailed in Methods.

Conclusions: 

Ln. 28 The general conclusions show that the Covid-19 pandemic…   Change to: ..general conclusion is… (one conclusion),  or results have shown…

Do you consider the topic original or relevant in the field, and if so, why? &   What does it add to the subject area compared with other published material?

The topic is relevant in the field of economy and work-related issues during the pandemic. The article present results of the research about how the pandemic affected the employee support system specifically in terms of health protection and quality of work. But it is primarily about work related issues, expectations form employees and perceived employers support, not public health or QOL.

Although the design, methods used and results are correct, this is not quite a public health approach, and the article does not correspond to the scope of the public health journal. The content does not correspond to this specific Section: Health-Related Quality of Life and Well-Being.

Since this is valuable work, the suggestion is to send the paper for consideration to a journal which has more economic and work-related issues (during pandemic period) in its scope.

Author Response

Thank you for taking the time to improve our manuscript. We have studied all comments and suggestions carefully. We tried to apply them. Below are our responses to each suggestion.

What is the main question addressed by the research?

The aim of the study is to investigate what forms of employees' support were expected and provided by employers during 19 the pandemic (in terms of health care and quality of work)

There are some suggestions to improve the text:

Ln 18. The aim of the study is to indicate what forms…  Instead of "indicate" use word to explore… or to investigate… to determine…  this is more acceptable in writing a scientific paper.

Ln 117-118 the aim was described somewhat better (the aim of the research is to determine how the pandemic affected the employee support system in terms of health protection and quality of work.) so suggestion is to put this formulation in Abstract and more detailed in Methods.

Answer:

Thank you for this suggestion. We corrected the text.

Conclusions:

Ln. 28 The general conclusions show that the Covid-19 pandemic…   Change to: ..general conclusion is… (one conclusion),  or results have shown…

Answer:

Thank you for this suggestion. We corrected the text.

Do you consider the topic original or relevant in the field, and if so, why? &   What does it add to the subject area compared with other published material?

The topic is relevant in the field of economy and work-related issues during the pandemic. The article present results of the research about how the pandemic affected the employee support system specifically in terms of health protection and quality of work. But it is primarily about work related issues, expectations form employees and perceived employers support, not public health or QOL.

Although the design, methods used and results are correct, this is not quite a public health approach, and the article does not correspond to the scope of the public health journal. The content does not correspond to this specific Section: Health-Related Quality of Life and Well-Being.

Since this is valuable work, the suggestion is to send the paper for consideration to a journal which has more economic and work-related issues (during pandemic period) in its scope.

Answer:

We submitted an article to a special issue of "Lifestyle, Nutrition, Perception of Health and Quality of Life and Care in Times of COVID-19". The editor of this issue has accepted our manuscript. The editor wrote, "Experimental studies and reviews of the effectiveness of coping strategies that contribute to improving the quality of life and mitigating the consequences of this situation in today's society are also welcome." We believe that our research fits within this framework.

Moreover, based on the definition of public health by A. Winslow (who was the first to define this concept), public health is both an area of ​​empirical knowledge - science - and a public activity with a clear system of values ​​- art. Its aim is to prevent disease, extend life and improve its quality. However, unlike medicine, it does not focus on treating diseases or injuries of individuals. On the other hand, it deals with preventing them in a broader dimension - social and environmental. In the case of our study, this social dimension is important. Besides, public health has traditionally been concerned with epidemics, disasters, hygiene and monitoring the health of the population. Nowadays, he also studies the factors influencing health and quality of life and deals with changing them for the better. It defines what a healthy lifestyle is about and promotes it. It shows the role of working or living conditions, public space and even culture. Thus, our study deals with the issue of working conditions.

In conclusion, our study is in line with the issues of the JERPH journal and the special issue we have selected.

At the same time, we thank you for appreciating our research by pointing out that it is valuable.

Reviewer 2 Report

Please note attachment. 

Author Response

Thank you for taking the time to improve our manuscript. We have studied all comments and suggestions carefully. We tried to apply them. In enclosure please find our responses to each suggestion.

Reviewer 2:

Thank you for taking the time to improve our manuscript. We have studied all comments and suggestions carefully. We tried to apply them. Below are our responses to each suggestion.

Introduction

1)  Lines 86-89: I can't believe it. Here, the current aspect of the gasoline costsshould

be evaluated.  The acquisition costs of digital media are also unique. And

companies that give out meal vouchers may also share in the acquisition costs.

Answer:

We also found the results of the indicated research original, which is why we have quoted them.

2)  Lines 90-93:  …technostress,…further remote work…:  What role does

technostress play here? Does technostress lead to more or less remote work?

Answer:

Technostress in the cited studies is treated as a determinant of the development of remote work. It is a destimulant. It is understood as a phenomenon (pointed out by the European Parliament) that determines the employee's feeling of discomfort related to the use of technology, i.e., specialized software and artificial intelligence, to supervise his work - remote monitoring of progress and performance in real time and tracking.

3)  Please explain the internet connection in Poland. Are there any differences

between urban and rural areas? Do they all can take online courses? Or those who

don't have the internet don't know internet and don't need it anyway, and therefore

don't need online courses?

10) What is the connection to the internet and thus the reachability of online courses

in rural areas?

Answer:

The differentiation between the countryside and the city in access to the Internet in Poland is negligible. While in large cities this indicator is 94.4%, in villages it reaches 93.2%. Generally, 93.3% had access to the Internet. households. Including fixed lines, broadband has 69.5 percent. households, and 70.6 percent. has access to mobile broadband internet. This data has hardly changed since the pandemic. This means that the conditions for remote work in Poland were very good.

Material and methods

4)  Lines 150 vs 153: Write numbers in a consistent way.

Answer:

Changes have been made to the text.

5)  What does the "Polish Classification of Activities" represent? What is the purpose

of people registering there?

Answer:

The Polish Classification of Activities is a division of economic activities carried out by economic entities in Poland. It was created for the purposes of statistics, records, accounting, documentation and use in official IT systems and registers. It includes everyone who starts businesses. It was developed on the basis of the NACE Rev2 statistical classification of economic activities, which was introduced by Regulation (EC) No. 1893/2006 of the European Parliament and of the Council of 20 December 2006 on the statistical classification of economic activities NACE Rev.2. Therefore, it is consistent with the classification NACE Rev.2.

To make it easier for the readers to understand what the Polish Classification of Activities is, we have provided an explanation in the text in parentheses.

6)  Was there approval by an ethics committee for you to access personal data? I

recommend adding the consent, because I'm not sure if there is a breach of EU

data protection law here (General Data Protection Regulation, GDPR). Or was the

"Polish Classification of Activities"created just for this survey?

Answer:

The institution that collected the data for us has an appointed data protection officer. This person represents the data protection organization. In the market research, the data protection officer was mentioned. Besides, the survey was anonymous. All invitations to fill out the questionnaires contained information about the possibility of opting out of responding to data protection. Thus, the study was conducted in accordance with Regulation (EU) 2016/679 of the European Parliament and of the Council of 27 April 2016 on the protection of individuals with regard to the processing of personal data and on the free movement of such data.

We added this remark to the research description.

7)  How exactly were the subjects invited?

8)  Describe the process of data collection in more detail.  Who conducted the

interviews? How many people were involved? What qualifications did these

people have. Was structured training provided beforehand? Were the interviews

recorded? If so, with what? Participation was certainly voluntar?

Answer:

The research was commissioned by a specialized research institution (outsourcing was applied). She was responsible for the whole organization of the research.

Results:

9)  What should say the variables P2. Unfortunately, I am not well versed in Poland.

Other readers perhaps also not. It would have been better to order by rural area,

small or big city. The results seem to me to be unnecessary.

Answer:

Voivodship - the highest-level administrative division in Poland. We were interested in whether there would be a differentiation of responses by voivodship. There are differences in Poland between the so-called Northern and southern Poland, western and eastern Poland, as in Germany or Italy. In our opinion, this variable has meaning and justification in our research.

We added this remark to the research description.

11) A person such as working in the mining industry will certainly be hard to

convince to follow an online course.

Answer:

In the case of the Mining and Quarrying sector, miners were not considered, only those working in managerial positions, in logistics or administration, in this case remote work may apply.

12) P4: Here I suspect that there is also more money in companies with > 500 people

and that the health program is better designed. The results confirm that health

protection in small companies is poorly trained. Do the very small and small

companies even have money to offer online courses?

13) P5: See comment above.

Answer:

The cost of organizing remote work depends on the type of activity, as well as the scope of remote work. So even in a very small company you can organize remote work at a low cost.

Discussion

14) The results should be discussed first, then the limitations should be pointed out. I

recommend exchanging paragraphs 5.1 and 5.2.

Answer:

Sections swapped as suggested.

15) Line 570: What is CSR?

Answer:

CSR - corporate social responsibility.

16) Overall, I think the results are realistic that there is currently no need for online

courses because the overall concept is not yet ready.

17) The hybrid form of work is a source of future - for most branches. Therefore, the

obligation to offer online courses also arises. However, it takes time and employee

awareness of these online courses. I am sure that the online courses will be better

accepted if they can be completed during working hours. It is very interesting to

know the course of this survey in 5 to 10 years.

Answer:

We agree with this opinion. In our opinion, the issue of remote work has great research potential. The pandemic has triggered the mechanism and a shift to remote or hybrid work can be expected in vulnerable sectors. Of course, there are types of activities that are definitely not susceptible to remote work.

Round 2

Reviewer 1 Report

Thank you for taking the time to improve your manuscript. After reading the additions as suggested by both reviewers, I have nothing more to add.

If the editor of the special issue accepts this topic as relevant to the special issue, I agree this version of the paper to be accepted for publishing in IJERPH.

Reviewer 2 Report

Thank you for the adaptation. All the best to you.